



# Geomorphology and shallow sub-sea floor structures underneath the Ekström Ice Shelf, Antarctica

Astrid Oetting[1,a], Emma C. Smith[1,b], Jan Erik Arndt[1], Boris Dorschel[1], Reinhard Drews[2], Todd A. Ehlers[2], Christoph Gaedicke[3], Coen Hofstede[1], Johann P. Klages[1], Gerhard Kuhn[1], Astrid Lambrecht[4], Andreas Läufer[3], Christoph Mayer[4], Ralf Tiedemann[1,5], Frank Wilhelms[1,6], and Olaf Eisen[1,5]

[1]Geosciences, Alfred Wegener Institute, Helmholtz Centre for Polar and Marine Research, Bremerhaven, Germany
[2]Department of Geoscience, University of Tübingen, Tübingen, Germany
[3]BGR, Federal Institute for Geoscience and Natural Resources, Hannover, Germany
[4]Geodesy and Glaciology, Bavarian Academy of Science and Humanities, Munich, Germany
[5]Department of Geoscience, University of Bremen, Bremen, Germany
[6]Department of Geoscience, University of Göttingen, Göttingen, Germany
[a]now at: Institut für Planetologie, Westfälische-Wilhelms Universität, Münster, Germany
[b]now at: School of Earth and Environment, University of Leeds, Leeds, UK

**Correspondence:** Emma C. Smith (E.C.Smith1@leeds.ac.uk)

**Abstract.** The Ekström Ice Shelf is one of numerous small ice shelves that fringe the coastline of western Dronning Maud Land, East Antarctica. Reconstructions of past ice-sheet extent in this area are poorly constrained, due to a lack of geomorphological evidence. Here, we present a compilation of geophysical surveys in front of and beneath the Ekström Ice Shelf, to identify and interpret evidence of past ice sheet flow, extent and retreat. The sea floor beneath the Ekström Ice Shelf is dominated by an

5   incised trough, which extends from the modern day grounding line on to the continental shelf. Our surveys show that Mega-Scale Glacial Lineations cover most of the mouth of this trough, terminating 11 km away from the continental shelf break, indicating the most recent maximal extent of grounded ice in this region. Beneath the front ~30 km of the ice shelf, the sea floor is characterised by an acoustically transparent sedimentary unit, up to 45 m-thick. This is likely composed of subglacial till, further corroborating the presence of past grounded ice cover. Further inland, the sea floor becomes rougher, interpreted

10  as a transition from subglacial tills to a crystalline bedrock, corresponding to the outcrop of the volcanic Explora Wedge at the sea floor.

Ice retreat in this region appears to have happened rapidly in the centre of the incised trough, evidenced by a lack of overprinting of the lineations at the trough mouth. At the margins of the trough uniformly spaced recessional moraines suggest ice retreated more gradually. We estimate the palaeo-ice thickness at the calving front around the Last Glacial Maximum to

15  have been at least 305 m to 320 m, based on the depth of iceberg ploughmarks within the trough and sea-level reconstructions.

Given the similarity of the numerous small ice shelves around the Dronning Maud Land coast, these findings are likely representative for other ice shelves in this region and provide essential boundary conditions for palaeo ice-sheet models in this severely understudied region.



## 1 Introduction

In the western Dronning Maud Land (wDML) sector of East Antarctica (see Fig. 1a), the area between the Riiser Larsen and Vigrid ice shelves, ice drains through numerous small ice shelves from a combined drainage area of ~470,000 km$^2$ (Rignot et al., 2013). Recent studies have shown that ice shelves in this region are underlain by deep glacial troughs (Nøst, 2004; Eisermann et al., 2020; Smith et al., 2020), indicating ice advanced towards outer continental shelf areas in the past. However, a lack of geomorphological evidence prevents the definition of the ice extent, flow regime, and pattern of ice retreat in the wDML region during the last glaciation. Consequently, the glacial history of this region is poorly constrained and limits our ability to understand past responses of this sector to climatic and oceanic variations. However, this is essential for evaluating future ice-sheet development in this sector of East Antarctica.

Mapping the continental shelf geomorphology around Antarctica with hydro-acoustic methods (swath bathymetry and sub-bottom profiling) is widely used to infer past ice-sheet evolution (e.g. Jakobsson et al., 2012; Lavoie et al., 2015; Arndt et al., 2017; Greenwood et al., 2018). For the wDML sector, however, such observations are largely absent. Systematic geomorphological surveys have been published in the neighbouring Coats Land region (Hodgson et al., 2018, 2019; Arndt et al., 2020), ~600 km to the west, and around Mac.Robertson Land (Mackintosh et al., 2011), ~2,900 km to the east. The lack of data in the wDML region is mainly due to limited accessibility. The continental shelf in this area is particularly narrow (Arndt et al., 2013) and is largely covered by thick ice shelves, preventing ship-based data acquisition. Autonomous underwater vehicles (AUVs) provide one solution to collect high resolution bathymetry data of sub-ice shelf sea floor structures (e.g. Jenkins et al., 2010; Davies et al., 2017). An AUV was used for exploring the cavity beneath Fimbul Ice Shelf (Nicholls et al., 2006) in wDML. However, no high resolution sea floor bathymetry data was collected since it was aimed to survey the bottom of the ice shelf. While AUVs are an effective technology for sub-ice shelf data acquisition, they are still cost-intensive and also carry a high risk of equipment loss, as documented by the loss of the AUV used under Fimbul Ice Shelf (Dowdeswell et al., 2008). Consequently, AUVs are still rarely used below ice shelves.

Here we present an alternative approach, for which we combined marine geophysical and on-ice reflection seismic data for the Ekström Ice Shelf (EIS) embayment, wDML (Fig. 1). Seawards of the ice shelf we use multibeam echosounder and parametric sub-bottom sediment echosounder (PARASOUND) data and on the ice shelf we use a grid of vibro-seismic reflection surveys. We provide a joint interpretation of these data sets to better resolve the poorly constrained ice-sheet dynamic history of the wDML, and moreover discuss advantages, disadvantages and limitations of this approach.

## 2 Study Area

Our study area covers the continental shelf and the sea floor under the front 50 km of the EIS cavity (Fig. 1). The EIS covers an area of 6800 km$^2$ (Neckel et al., 2012) and drains into the Atlantic Sector of the Southern Ocean. It is bounded by the ice



rises of Sørasen to the west and Halvfarryggen to the east. Geophysical mapping of the sea floor bathymetry beneath the ice
shelf (Eisermann et al., 2020; Smith et al., 2020) revealed an incised trough extending around 140 km from the most landward
location of the current grounding line (8°6'W at the southernmost point) to the continental shelf. The distance to the grounding
line is measured along the trough axis. The trough has a maximum depth of 1100 m below sea level, 30 km downstream (north)
of the current grounding line (Fig. 1b) and shallows to ~550 m on the continental shelf (Arndt et al., 2013). Smith et al. (2020)

concluded a palaeo-ice stream extent towards the continental shelf edge near the current ice-shelf front from where it retreated
episodically, indicated by topographic highs along the trough axis. Sediment core evidence offshore the EIS further suggests
an extended East Antarctic Ice Sheet during marine isotope stages 6 and 4-2 probably reaching beyond the continental shelf
break (Grobe and Mackensen, 1992), but no direct geomorphological or geological evidence exists for that so far (Hillenbrand
et al., 2014).

Schannwell et al. (2020), using the bathymetry of Smith et al. (2020); Eisermann et al. (2020), modelled the evolution of the
ice in the Ekström Ice Shelf embayment over a 40,000 year glacial cycle, investigating the effect of differing bed materials on
ice dynamics. They concluded that the ice reached its maximum extent ~20,000 years before present during the Last Glacial
Maximum (LGM: 23–19 cal. ka BP). Significantly, the maximum ice extent was simulated to be located ~50 km inland of
the continental shelf break. Furthermore, this maximum ice extent simulated by Schannwell et al. (2020) was only reached if

ice was characterised by a "hard, high-friction" bed. By using a "soft, low friction" bed as boundary condition, the maximum
ice extent was ~100 km inland of the continental shelf break (Schannwell et al., 2020). In conclusion, the current range of
estimates of maximal ice-sheet extent differs significantly, highlighting the uncertainty about the glacial history of this region
and the factors affecting ice advance and retreat.

## 3  Data Sets and Methods

We investigated the sea floor in front of the ice shelf with marine geophysical methods and used reflection seismics to image
areas below the EIS (see Fig. 2).

### 3.1  Marine Geophysical Methods

Bathymetric data used in this study were collected during 16 expeditions of the RV *Polarstern* (Alfred-Wegener-Institut
Helmholtz-Zentrum für Polar- und Meeresforschung, 2017) between 1989 and 2016 and are available at PANGAEA (see sup-

plementary information S1). The data were recorded with the hull mounted multibeam echosounder systems Atlas Hydrosweep
HSDS 2 and HSDS 3 (Knust, 2017). All multibeam echosounder data were processed with the hydrographic software suite
HIPS & SIPS from Teledyne CARIS. In this process, all data were corrected for sound velocity variations in the water column
and erroneous measurements were removed. Processed multibeam data were exported to ASCII files and subsequently grid-
ded at 25 m resolution using a weighted moving average gridding algorithm in the QPS Fledermaus software. Details on the

multibeam data and their processing can be found in Arndt et al. (2013).





The sub-bottom profiler data used in this study were recorded during expeditions PS82 in 2014 (Damaske and Kuhn, 2014) and PS96 in 2015/2016 (Arndt and Kuhn, 2016) of the RV *Polarstern* using a hull-mounted Atlas Parasound P70 sediment echosounder system (Knust, 2017). The Parasound device was operated with primary frequencies of 18 kHz and 22 kHz, resulting in a secondary frequency of 4 kHz. The master tracks of these RV *Polarstern* expeditions are shown in Figure 1b.

## 3.2   On-Ice Reflection Seismics

Data beneath the ice shelf were acquired using an on-ice vibroseis source. The source signal is a linear sweep lasting 10 s with a frequency of 10 Hz to 220 Hz (Smith et al., 2020). A snow streamer was used to receive the reflected seismic signals. It contained 60 channels, each channel consisted of a group of eight geophones (Eisen et al., 2015; Smith et al., 2020). The seismic profiles were acquired during the Antarctic summer seasons 2016/17 and 2017/18 (hereinafter referred to as the 2017

and 2018 data sets, respectively). The layout of the seismic profiles is shown in Figure 1b.
Further technical details regarding the vibroseis method are given in Eisen et al. (2015). Details of acquisition on the Ekström Ice Shelf are given in Smith et al. (2020), who studied the cavity geometry below the ice shelf but did not address geomorphological landforms. The vertical resolution for the stacked data at the sea floor is ~6 m. The horizontal resolution ranges between 50 m and 130 m, depending on the shot point distance. The uncertainty in sea floor depth determined from these data sets is

14.8 m (Smith et al., 2020).

The horizontal and vertical resolution of seismic data is, as described above, lower than that of the marine geophysical data, and the spatial coverage much more sparse. Accordingly, we take care interpreting these data, as features are often imaged by a single seismic line only, limiting our ability to confirm the orientation or spatial extent of features. An evaluation of the seismic data is given in Section 5.2.

## 4   Results and Interpretation

We describe landforms identified in front of and beneath the modern day EIS, compare them to similar landforms found on other formerly glaciated continental shelves, and discuss later potential formation processes. Due to differences in data density and resolution between the marine and the on-ice reflection seismic data, we discuss the results separately in terms of the outer continental shelf (Section 4.1) and the sub-ice shelf (Section 4.2).





### 4.1 Marine Geophysical Data: The Outer Continental Shelf

#### 4.1.1 Iceberg Ploughmarks

The outer continental shelf is covered by a multitude of curvilinear, elongated furrows (Fig. 3a, marked in yellow). Near the continental shelf break, they occur in water depths down to 535 m. At the eastern margin of the trough they occur in shallower water depths of about 320 m. In the deeper parts of the trough, near the ice shelf front, the furrows are only present at water depths shallower than 420 m. These furrows are randomly oriented and occasionally cross-cutting. They are several 100s m to more than 1000 m long, have widths ranging from a few meters (visible in the PARASOUND echograms) up to 100 m (Fig. 3b), and are up to 10 m deep. Based on the furrow characteristics, we interpret these as iceberg ploughmarks formed by the keels of drifting icebergs that have grounded on the sea floor, reworking the upper sea floor sediments along its path. Iceberg ploughmarks are a common feature of polar continental shelves and have also been identified a few hundred kilometers southwest of the EIS in the southern Weddell Sea (Lien et al., 1989; Gales et al., 2016; Arndt et al., 2017, 2020).

#### 4.1.2 Mega-Scale Glacial Lineations

The deepest part of the trough near the modern calving line is characterised by sets of parallel to sub-parallel linear ridges (Fig. 3a and c, blue lines). These ridges are between 1 km and 8 km long and have widths between 150 m and 450 m, resulting in elongation ratios between 15:1 and 40:1. The mean spacing between ridge crests is 250 m. The ridges are observed in water depths between 380 m and 550 m with amplitudes ranging from less than one meter up to 10 m. The ridges strike northwestwards aligning with the trough axis. Closer to the trough edges, a divergent orientation of the lineations is observed, at the eastern edge of the trough they have a direction of 155° from N, at the western edge of the trough mouth they are 115° from N orientated (see Fig. 3a).

Sub-bottom profiler data acquired across these ridges indicate that they were moulded into an acoustically transparent layer (Fig. 3e). A shallow and continuous sub-bottom reflector is visible beneath that transparent layer, which fades out further southwestwards. It is likely that this continuous reflector is only visible beneath the smaller ridges, where the acoustically transparent layer is thinner and the sub-bottom profiler signal can penetrate deep enough to resolve it.

We interpret those ridges as mega-scale glacial lineations (MSGLs). MSGLs are widely documented in former glaciated regions and have been described on land (Clark, 1993), on the sea floor (e.g. Ó Cofaigh et al., 2002), and beneath active ice streams (King et al., 2009). Their occurrence indicates the former presence of a fast-flowing ice stream (e.g. Stokes and Clark, 1999; Livingstone et al., 2012). The acoustically transparent layer into which the MSGLs are formed is often associated with a soft deformable till of varying shear strengths (Dowdeswell et al., 2004; Ó Cofaigh et al., 2005). Their slightly varying orientation in front of Ekström Ice Shelf indicates a diverging palaeo-ice flow on the outer continental shelf.





### 4.1.3 Recessional Moraines

Close to the modern-day calving front, a set of linear and sub-parallel ridges is located at the eastern trough margin (Fig. 3a, turquoise lines). The mostly symmtrical ridges are up to 15 m high and have lengths between 1 km and 5.7 km (Fig. 3d). Their widths vary between 100 m and 300 m. The ridges occur at a relatively uniform spacing with a distance from crest to crest of about 300 m. The orientation of the ridges is approximately perpendicular to the trough axis. Further to the trough centre their orientation changes to 85° in clockwise direction. The water depths in the area where the ridges occur range between 390 m
and 445 m.

Ridges of similar shapes and dimensions were found in the centre of formerly glaciated cross-shelf troughs and have been interpreted as recessional moraines, e.g. in the Amundsen Sea Embayment (e.g. Klages et al., 2015), in the Ross Sea (e.g. Simkins et al., 2016) and offshore northeast Greenland (e.g. Winkelmann et al., 2010). Accordingly, we interpret these linear and sub-parallel ridges as recessional moraines. Such sets of recessional moraines form at the ice margin by pushing and
squeezing of sub- and proglacial deposits during short halts or minor readvances of the ice terminus during general ice-sheet retreat (e.g. Winkelmann et al., 2010).

### 4.2 On-ice Seismic Reflection Data: Sub-Ice Shelf

This study focuses on shallow sub-sea floor features, in the upper ~100 m of the sea floor. We do, however, map one deeper geological feature, the volcanic Explora Wedge (Kristoffersen et al., 2014), where it outcrops at the sea floor (Fig. 4a, red line).
The Explora Wedge divides the study area into two regions, marking a transition between soft and hard sea floor material. Accordingly, the landforms discussed here are divided into two groups, those downstream (north) (Section 4.2.1) and those upstream (south) (Section 4.2.2) of the outcropping Explora Wedge.

### 4.2.1 Landforms downstream of the Explora Wedge outcrop – Smoother sea floor

The seismic data collected downstream (north) the outcropping Explora Wedge mainly cover the eastern part of the trough
(Fig. 4 and 5). This area is shallower than the centre of the trough with water depths ranging from 250 m at the trough edge to 550 m closer to the trough centre and is mostly flat. Therefore, we refer to this area as the plateau hereafter. The seismic data show that the sea floor of the plateau is relatively smooth, with a clear and continuous sea floor reflection. The majority of the plateau is covered by a seismic unit with its base characterised by a clear sub-bottom reflector. This unit is between 13 m and 45 m thick, overlying truncated dipping reflectors (Fig. 4g-i, purple). The thickest part of this sedimentary unit is located in a
depression south-west of the German Neumayer III station (Fig. 4a, black dashed line) that is 16.3 km long (measured along seismic line 20170552) and 12.6 km wide (measured along seismic line 20180551).

Within this seismic unit we have identified five distinct thicker protrusions at the sea floor, from here on we will refer to these as "bumps" (Fig. 4a-f). The bumps in Figure 4b, c and e are intersected by seismic lines in an along-flow direction. The bumps in Figure 4b and c have thicknesses of around 20 m and along-flow extents of 1200 m and 1800 m respectively. They have
asymmetric forms with a steeper flank on the ice-faced side and a shallower flank in the ice-distal side direction. The bump in



Figure 4e is the thickest feature at 40 m and extends in an along-flow direction for 820 m, also showing a slightly asymmetrical form. Bumps in Figure 4d and f are sampled by seismic lines oriented in an across-flow direction, both are around 15 m thick and extend for 1.5 km and 390 m respectively. They both have relatively symmetrical forms.

On the western edge of the plateau, as the sea floor begins to deepen into the trough, a distinct layered unit is imaged
across a number of seismic lines (Fig. 5b-f), both in the along-flow and across-flow directions. This unit, once again, overlies truncated dipping layers and appears to pinch out against these layer in shallower water depths at around 540 m (Fig. 5b). It is imaged down to depths of 800 m (Fig. 5d). The extent over which the feature could be mapped is shown in Figure 5a. The unit comprises of two clearly defined layers. The uppermost of the two layers is thicker, reaching a maximum thickness of around 40 m, close to the current calving front (Fig. 5a-f, light pink) and has a greater spatial extent. The lower layer has a maximum
thickness of around 25 m, and is less extensive (Fig. 5a-f, dark pink). The maximum thickness of the whole unit is 65 m.

### 4.2.2 Landforms upstream of the Explora Wedge outcrop – Rougher sea floor

Upstream of the Explora Wedge outcrop, the sea floor is qualitatively rougher than downstream of the outcrop (Fig. 6a-d). In the centre of the incised trough, at a water depth of around 900 m, we image a series of hummocky landforms in the along-flow direction and on a gently curving seismic line, at approximately 25° to the along-flow direction (Fig. 6a, b). Some of the
landforms have steeper sides in the upstream-facing direction and more gently sloping sides in the downstream-facing direction (Fig. 6b and c, red arrows), although it is not simple to divide them into individual features. They range in height from 30 m and 90 m, measured from the most elevated point of the landform to the surrounding sea floor. Some of these landforms show morphological lows on their ice-faced side (Fig. 6b, c).

In the across-flow direction, directly upstream of the volcanic Explora Wedge outcrop, we image U-shaped depressions at
the base of the trough flanks (Fig. 6e, f), with widths between 3 km and 5 km and depths of 20 m to 100 m compared to the average surrounding sea floor.

## 5 Discussion

### 5.1 Palaeo-ice sheet setting from marine geophysical evidence

An overdeepened trough is present beneath the EIS, which widens and shallows close towards the modern ice shelf front (Smith
et al., 2020; Eisermann et al., 2020). Such troughs are common on Antarctic shelves (e.g. Ó Cofaigh et al., 2002; Livingstone et al., 2012; Graham et al., 2016a, b; Arndt et al., 2017; Larter et al., 2019) and are interpreted as the conduits of palaeo-ice streams during glaciations. Similar incised troughs are also present at adjacent ice shelves (Eisermann et al., 2020; Nøst, 2004; Favier et al., 2016). Smith et al. (2020) concluded that this trough was the location of a palaeo-ice stream, we corroborate and extend this interpretation with addition of geomorphological evidence. The presence of MSGLs on the continental shelf,
terminating 11 km from the continental shelf break (Fig. 3a), are evidence that fast flowing ice covered this area in the past. Fanning of the MSGLs at the trough mouth suggests divergent ice flow unconstrained by topography, typical of ice margins.





Further evidence that grounded ice was present close to the continental shelf break, is the presence of recessional moraines at the trough edges, perpendicular to the paleo-ice flow direction indicated by the MSGLs (Fig. 3a, b).

The age of MSGLs and recessional moraines in the EIS embayment is difficult to constrain with currently available data. Grobe and Mackensen (1992) investigated sediments offshore the EIS, which did not yield any datable material to constrain a minimum age for ice sheet retreat. In addition, our data reveal that the sediment core location PS1385 is probably strongly affected by iceberg ploughing (Fig. 1, pink dot) likely obliterating any stratigraphic successions. Other cores recovered from the continental slope suggest that grounded ice may have reached the shelf break during past glaciations, including the LGM (Grobe and Mackensen, 1992). The thin drape layer on top of the MSGLs (Fig. 3e), and the fresh appearance of MSGLs and recessional moraines in the multibeam data, fare evidence that these features are relatively young, i.e. LGM and or younger.

The moraines at the trough edges (Fig. 3a, b) are the only landforms indicative of short-lived grounding-line stillstands during ice retreat. It is possible such features were also present in the trough centre, however this area is heavily covered by iceberg ploughmarks, which may have eradicated those moraines in the deeper parts of the trough.

The iceberg ploughmarks (Section 4.1.1) which cover large parts of the outer continental shelf (Fig. 3d) reach greater depths than ploughmarks within Ekström embayment, indicating that the icebergs creating these two sets of ploughmarks have different origins. The mainly east-west oriented ploughmarks at and beyond the continental shelf break were very likely formed by large icebergs transported from somewhere east of EIS and driven predominantly by currents (e.g. Antarctic Coastal Current) and wind over longer distances (Stuart and Long, 2011).

A bathymetric sill is present at the continental shelf break, separating the open ocean from the over-deepened incised trough, at it's deepest point, the sill is at approximately 390 m water depth (Arndt et al., 2013). Therefore, icebergs calved outside Ekström trough were only able to enter the trough with a draft of less than 390 m modern day water depth. It is, therefore, reasonable to assume that ploughmarks deeper than 390 m within the Ekström trough can only have been calved from the EIS itself. The ice shelf thickness of 2017 at the calving front (line 20170551) is approximately 250 m, thus making the draft of the produced icebergs too shallow to produce ploughmaks of that dimension. Therefore, the ploughmarks at water depths of up to 420 m within the Ekström trough, indicate that the ice shelf was thicker at some time since recession of the calving line beyond the iceberg ploughmark locations and that the maximum ploughmark water depth can be used to estimate the maximum ice-shelf thickness during this time (see Section 6).

## 5.2 Interpretation of sub-ice shelf seismic data

The on-ice reflection seismic data extend the marine geophysical coverage and image the continuation of the trough beneath the ice shelf. Due to limitations in the seismic data (see Section 5.2) we interpret these features in context of the ice dynamic setting laid out by the marine geophysical data, which gives us a reference frame to constrain the types of landforms that may be expected in this realm. Furthermore, the seismic data allowed us to broadly identify a divide between qualitatively smoother or rougher sea floor, separated by the outcrop of the Explora Wedge at the sea floor. The rougher sea floor upstream of this wedge is likely made of more resistant crystalline bedrock, whereas the area downstream is likely covered with sedimentary substrate. A similar setting is observed in many other areas of Antarctica such as Pine Island Bay (Nitsche et al., 2013; Graham





et al., 2016a), Marguerite Bay (Livingstone et al., 2013) or the Antarctic Peninsula (Larter et al., 2019). This transition in basal properties would have affected the palaeo-thermal regime and ice-flow regime, as ice-stream flow accelerates at the transition from crystalline to sedimentary substrate (e.g. Wellner et al., 2001; Ó Cofaigh et al., 2002; Lowe and Anderson, 2003; Siegert et al., 2005; Benn and Evans, 2010). Therefore, it provides evidence for the further interpretation of landforms identified in the
seismic data.

**Subglacial Till**

We interpret the 13 to 45 m thick seismic unit that overlays truncated dipping reflectors across much of the outer ice shelf cavity (Fig. 4), as a subglacial till. This sedimentary material is commonly observed below ice sheets and glaciers. For example, studies on the Marguerite Trough show sediment thicknesses reaching 15 m (Ó Cofaigh et al., 2002) and 20 m (Dowdeswell
et al., 2004), both similar to the thicknesses seen in our study. Basal moraines may show MSGLs (e.g. Ó Cofaigh et al. (2002); Dowdeswell et al. (2004)), however, the resolution and line spacing of the seismic data used in this study would not be sufficient to image MSGLs of the dimensions seen in front of the ice shelf. Therefore, and we cannot definitively say if they are present here.

       Downstream (north) of the outcropping Explora wedge, on the sub-ice shelf trough margin, a horizontal layered deposit is
present at the sea floor (Fig. 5b-f). It has two clear layers, which can be distinguished from each other (overview in Fig. 5a and e-f, pink and dark pink). As described in section 4.2.1, the top layer is thicker than the bottom layer and covers a larger spatial extent, pinching out against the trough flanks. An interpretation as subglacial till would also be reasonable in this area, since the location and sediment thickness are consistent with previous findings of such features (e.g. Engelhardt and Kamb, 1997; Tulaczyk et al., 1998; Ó Cofaigh et al., 2002; Dowdeswell et al., 2004; Epshtein, 2017; Lowe and Anderson, 2003).
The two distinct layers could be the result of a rapid ice sheet retreat, without major erosion, preserving the bottom layer and subsequent re-advance, forming the upper layer. This would also fit with the well-preserved MSGLs imaged in the marine geophysical data on the continental shelf. Althought it should be acknowledged that the fresh-looking MSGLs could be due to the removal of loose sediments by water flow beneath the ice shelf (Klages et al., 2017).

       An indication that periods of retreat and re-advance are possible in the EIS embayment is given by Schannwell et al. (2020),
who simulated an idealised advance and retreat history of the grounding line of the EIS embayment over the past 40,000 years. The simulations used a variety of bed conditions and showed that with both hard and soft beds, there were three periods of grounding line stillstand during advance and one in the retreat phase. A re-advance could be possible after such stillstands. Furthermore, the situation is likely more complex that that simulated by Schannwell et al. (2020), as our investigations have revealed a change in bed conditions 150 km from the grounding line, rather than a purely hard or soft bed, which has not been
considered in the simulations so far. An alternative scenario, although less likely, is that the lower layer originates from an earlier glaciation. However, this interpretation will remain ambiguous without age estimates from cores.





**Bumps in Subglacial Till**

The landforms we have called "bumps" (Fig. 4b-f) occur in an area of the plateau where sub-glacial till is present. The asymmetric form of the bumps imaged by seismic lines along the ice flow direction with a steep flank facing the ice and a shallower
ice-distal flank (Fig. 4b, c, e), as well as their dimensions to some extent, resemble the shape of drumlin or drumlinoid landforms. These landforms are generated under fast-flowing ice conditions below glaciers or ice streams by erosion, mobilization or deposition of sediment (e.g. Clark, 1993; Smith et al., 2007; Graham et al., 2009). The more symmetrical shape of the bumps revealed by the across-flow profiles (Fig. 4d, f) may support an interpretation as drumlins. If we assume the along-flow and across-flow lines are cutting a family of such features orthogonally, then the heights and lengths of these features are
within typical ranges for drumlinoids (Clark et al., 2009). The feature in Figure 4f is within the range for the typical width of a drumlinoid, however the feature in Figure 4d is wider than an expected width of 1500 m.

The plateau area is outside of the main trough, the latter being where we would expect the thickest and fastest flowing ice to have been. However, the plateau is largely at a similar depth to the area of sea floor on the continental shelf, where MSGLs have been identified from marine geophysical data. MSGLs provide evidence for fast paleo-ice flow velocities (Section 4.1.2)
and suggest fast ice flow was possible in these shallower areas. We also interpret the plateau area to be covered by subglacial till (Section 4.2.1), providing a source of material for drumlin formation. The presence of MSGLs already implies the presence of a till.

Drumlinoid landforms in the Western Amundsen Sea Embayment (Klages et al., 2015) were found to be preferably formed at the transition between soft and more resistant bedrock (Wellner et al., 2001) and the bumps we observe in our survey are
around 15 to 20 km downstream of the proposed bed-material divide of the Explora Wedge outcrop. Based on the limited seismic data coverage an unambiguous classification of these features as drumlinoids cannot be made at this stage, however, we conclude that these features are likely to be formed by subglacial sediment moulded by fast flowing ice.

**Hummocky Landforms in Bedrock**

An unambiguous interpretation of the hummocky landforms, identified in the centre of the sub-glacial trough upstream of the
Explora Wedge outcrop was not possible. However, some of these landforms appear to have an asymmetrical form with steeper ice-facing-side (see Fig. 6a-d). This shape may be comparable to landforms formed in resistant bedrock by continuous erosion by ice streaming, e.g. crag-and-tails or whalebacks. Crag-and-tails have been found around Antarctica mainly occurring in the centre of sub-ice shelf troughs in resistant bedrock, e.g. by Graham et al. (2009) in the Getz A/Dotson trough and Nitsche et al. (2013) in Pine Island Bay. Whalebacks have been identified for example in crystalline bedrock in the inner- or midshelf
of Marguerite Bay, Antarctica (Livingstone et al., 2013), and in British Columbia, Canada (Evans, 1996). These landforms are also formed in fast ice-flow environments (Evans, 1996; Roberts and Long, 2005). While the setting of these landforms is consistent with the setting inferred for the region in which the hummocky landforms occur at the EIS, meaning harder bed material upstream of the Explora Wedge and ice stream activity, and the general shape may be roughly similar to crag-and-tails or whalebacks, there is insufficient evidence to classify these landforms unambiguously as such.





## U-Shaped Depressions

Meltwater channels at the trough margin are expressed by U-shaped depressions (Fig. 6a, e, f), with dimensions comparable to meltwater channels (300-500 m wide, 30-200 m deep, Livingstone et al. (2013)) discovered in Marguerite Bayand Pine Island Bay (1-2 km wide, 200-400m deep, Nitsche et al. (2013)). These channels are interpreted to have formed by meltwater erosion into hard substrate at the ice-sheet base (e.g. Wellner et al., 2001). The depressions cannot be tracked over a larger area in our data set, why their interpretation as meltwater channels remains tentative and needs to be confirmed by more detailed surveys. However, the absence of meltwater flow paths in the swath bathymetry data might be also a result of a diffuse flow in subglacial till, which was already assumed in the Marguerite Trough (Dowdeswell et al., 2004). Another possible explanation might be that these deepenings did not originate from meltwater discharge, but from erosion of softer material by the ice-stream base, probably along tectonically derived pre-existing fault lines.

## Assessment of sub-ice shelf seismic data for the interpretation of glacial landforms

The on-ice reflection seismic surveys on EIS provide crucial data on the previously unknown bed character, sub-bottom geometry and sea floor features. Our geomorphological interpretation infers a range of landforms and sub-bottom structures that provide insights into the glacial history of this area. However, it also demonstrates the uncertainties this method has for revealing glacial geomorphological structures. The seismic data proved to be especially capable of revealing sub-sea floor features. For example, they allowed us to reveal that a large area is covered by two seismic units that we interpreted as subglacial till on top of dipping sediments, that likely were truncated by glacial overriding. In addition, the seismic data were capable of showing differences in the sea floor roughness beneath the ice shelf, which, according to numerical ice-flow modelling, was important for the Ekström paleo-ice stream system (Schannwell et al., 2020).

The capability of this methodology for classical sea floor geomorphology interpretation, however, is limited due to the low data density, especially in direct comparison to swath bathymetry data. The majority of the smaller landforms imaged are crossed by only one seismic line either across or along the predominant ice-flow direction. Therefore, it is not possible to determine the exact 3D shape of these landforms and neither their orientation or their extent could be finally resolved. In our case, this short-coming could be overcome to some extent by the availability of marine geophysical data directly offshore the calving front. These data provided a reference setting for the ice shelf cavity by verifying the presence of a palaeo-ice stream in Ekström Trough through the mapping of MSGLs. Landform classification and mapping with on-ice seismic data cannot be done to the same level of certainty as with marine geophysical methods which have the advantage of a much higher resolution, capable of detecting also small-scale landforms.

The seismic data used here were originally acquired to image deeper geological stratigraphy and their use in this near-surface sea floor study was somewhat opportunistic. In the future the use of on-ice seismic surveying for similar studies should not be ruled out. However, it certainly is more effective if combined with higher resolution geomorphological mapping, e.g. using vessels in front of the ice shelves and at least some AUV dives in the cavity if possible. This would also enable seismic surveys to be targeted at features of interest, where the deeper penetration of seismic imaging would be of particular use.



Nevertheless, the collection of on-ice seismic data proved to be cheaper methodology to collect at least basic sea floor depth information in comparison to using an AUV, which in addition is at risk of being lost during a dive. Therefore, we conclude
that this methodology is worth to be applied and developed further to extend our understanding of the most inaccessible and least surveyed parts of Antarctica. This is especially true for the Dronning Maud Land, where the continental shelf is narrow and nearly entirely covered by ice shelves, strongly limiting marine geophysical data acquisition.

**Implications for future investigations**

A general challenge of the Antarctic geoscience community is to quantify the LGM ice mass budget and post-LGM mass loss,
which reveals past ice sheet variability and the Antarctic Ice Sheet contribution to post-LGM sea level rise. Our findings here can be used to determine and evaluate boundary conditions for palaeo-ice flow models of this wider region in this relatively poorly-investigated sector of East Antarctica, allowing us to understand and simulate future behaviour of the ice shelves more reliably, e.g. as facilitated by the recently launched SCAR Scientific Research Program INSTANT. Our results contribute to the solution of those challenges along two lines. Firstly, we provide new observational evidence. The EIS is a typical ice shelf of
the wDML region (Neckel et al., 2012). Neighbouring ice shelves have been shown to be underlain by similar incised troughs (Eisermann et al., 2020; Nøst, 2004; Favier et al., 2016), and are expected to behave similarly and exhibit comparable palaeo-ice thickness, retreat and advance styles. Relatively warm Circumpolar Deep Water masses have a significant influence on ice shelf variability (e.g. Jacobs et al., 1996, 2011; Hillenbrand et al., 2017; Paolo et al., 2015; Hattermann, 2018). However, as assumed for the neighbouring Fimbul Ice Shelf (Hattermann et al., 2014) as well as for the EIS (Smith et al., 2020), these warm
water masses enter the cavity below the ice shelf only in relatively small amounts. Further evidence for the variability in the geological past with samples is aimed to be collected during the *Polarstern* expedition EASI-1 starting next year. Secondly, we combined results from different geophysical observation methodologies in open water and underneath the ice shelf. Although observations from underneath a single ice shelf with reflection seismic approaches have their limitations (see Section 5.2), they nevertheless provide both, insights into the sea floor properties and stratigraphy as well as valuable complementary information
about sea floor geomorphology. Future reflection seismic deployment, e.g. as envisaged for the pre-site surveys of the SWAIS-2C drilling (Patterson et al.) underneath the Ross Ice Shelf, can thus be optimised to not only profile sub-sea floor stratigraphy, but also consider better retrieval of geomorphology. At the same time, we showed the limitations of that approach, thus avoiding any false expectations.

## 6 Conclusions

We have identified a diverse range of landforms beneath and in front of the EIS using a combination of geophysical methods: swath bathymetry, sub-bottom profiler and seismic reflection data. The survey area is characterised by a deep trough running from the modern-day grounding line under the ice shelf and on to the continental shelf in front. At the trough mouth we have identified MSGLs, confirming that this trough was likely occupied by a palaeo-ice stream. The MSGLs stop around 11 km from the continental shelf break, which suggests this was most likely the maximum grounded ice extent during the most recent



glaciation. We therefore present the first geomorphological evidence for LGM ice sheet advance reaching the continental shelf break in this sector of Antarctica. Beneath the ice shelf, a sediment unit up to 45 m thick, covering much of the survey area and interpreted as subglacial till, adds evidence that this area was indeed covered by ice for some considerable time.

We have found sparse evidence to classify the ice retreat style in this area but a lack of overprinting of the MSGLs suggests the ice initially retreated rapidly from the outer shelf. At the eastern edge of the survey area, where the sea floor is shallow and thus palaeo-ice thickness was likely less, we have evidence of a slower, periodic ice retreat, recorded by a series of moraines. Further inland, it was not possible to robustly classify glacial landforms using on-ice reflection seismic data and thus the ice retreat style further inland remains unclear. However, we do note a transition between smoother and rougher sea floor, which will have influenced palaeo-ice dynamics in the area, since ice flow velocities are higher on sedimentary substrate than on crystalline bedrock (e.g. Wellner et al., 2001; Ó Cofaigh et al., 2002; Lowe and Anderson, 2003; Siegert et al., 2005; Benn and Evans, 2010). Depressions are present at the trough margins, which potentially suggesting more intense meltwater flow at some point in the past. Since there is no indication of meltwater discharge in the swath bathymetry data, it is possible, as already assumed in the Marguerite Trough, that the water follows a diffuse path in the subglacial till (Dowdeswell et al., 2004).

Based in iceberg ploughmarks and sea-level reconstructions we estimate that the ice thickness at the calving front in the Ekström Trough was ~305 m to 320 m at the time of calving between LGM and 10 ka BP (see supplementary material S2 for calculations).

Methodologically, we also tested the approach of using seismic reflection data in combination with marine geophysical data, to obtain information about the sea floor and near-surface sea floor beneath the EIS. Our results show that the combined methods allow a fairly good classification of the landforms in their glacio-geological context and give important geomorphological information, such as the presence of medium- to large-scale landforms, hard or soft bedrock and sea bed roughness. Hence, these methods are suitable to provide crucial constraints for palaeo-ice flow models and the evaluation of their performances.

*Data availability.* Links to data repositories will be supplied upon acceptance of the manuscript.

*Author contributions.* All authors contributed to discussion of the manuscript, data interpretation, and contributed comments toward the final version. AO designed and wrote the bulk of the paper, mapped geomorphological features and interpreted these; ECS, OE and JEA contributed text to sections of the paper and edited the manuscript; ECS additionally performed seismic data acquisition, processing and mapping of the volcanic Explora Wedge; A Lambrecht and CM performed seismic data acquisition; CH was in charge of the seismic equipment; OE coordinated and implemented the seismic field work; JEA and BD lead the acquisition and processing of marine geophysical data and together with JPK interpreted the data; RD, TAE provided discussion on data interpretation based on regional ice-flow modelling results; GK, CG, A Läufer, RT, FW and OE are Co-PIs on the project that supplied the seismic data.

*Competing interests.* Olaf Eisen was co-chief of this journal until April 2021 and is now editor.



*Acknowledgements.* All data were collected and provided by the Alfred Wegener Institute, Helmholtz Centre for Polar and Marine Research (AWI) and campaign partners, Federal Institute for Geoscience and Natural Resources (Bundesanstalt für Geowissenschaften und Rohstoffe, BGR) and Bavarian Academy of Science and Humanities, Commission of Geology. The seismic profiles were carried out as part of the 'Sub-Ekström Ice Shelf observation' (Sub-EIS-Obs) capaign in 2016/17 and 2017/18. The data processing by ECS were funded through the AWI-BGR Sub-EIS-Obs Project and the DFG Cost S2S project Grant EI672/10-1 in the framework of the priority program "Antarctic Research
with comparative investigations in Arctic ice areas". RD was supported by an Emmy Noether Grant of the Deutsche Forschungsgemeinschaft (DR 822/3-1).



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



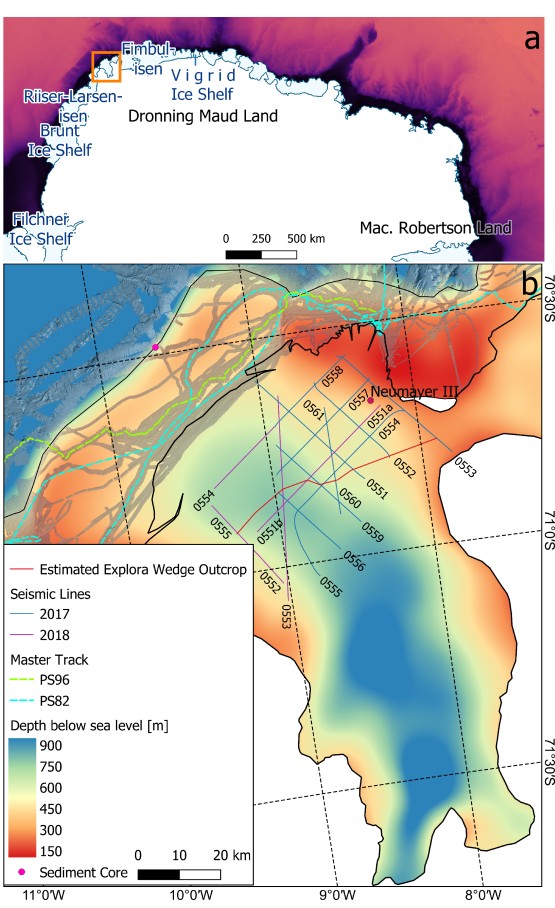

**Figure 1.** a) Overview of Dronning Maud Land within Antarctica, study area is highlighted with orange box (b) Map of the study area, in front of and beneath Ekström Ice Shelf. Ice shelf: sea floor depth is from Eisermann et al. (2020), seismic lines used in this study are indicated by the purple and blue lines and the line numbers are indicated. The outcrop of the volcanic Explora Wedge (Kristoffersen et al., 2014) at the sea floor is shown as a red line and is based on an interpretation of the seismic profiles detailed here. In front of the ice shelf: colour coded high-resolution swath bathymetry data gives the sea floor depth, Parasound-tracking lines from Knust (2014); Schröder (2016) are shown as blue and yellow dashed lines. The grey contour line shows the 1000 m below sea level depth. The pink dot on this contour line shows the location where Grobe and Mackensen (1992) sampled sediment core PS1385.



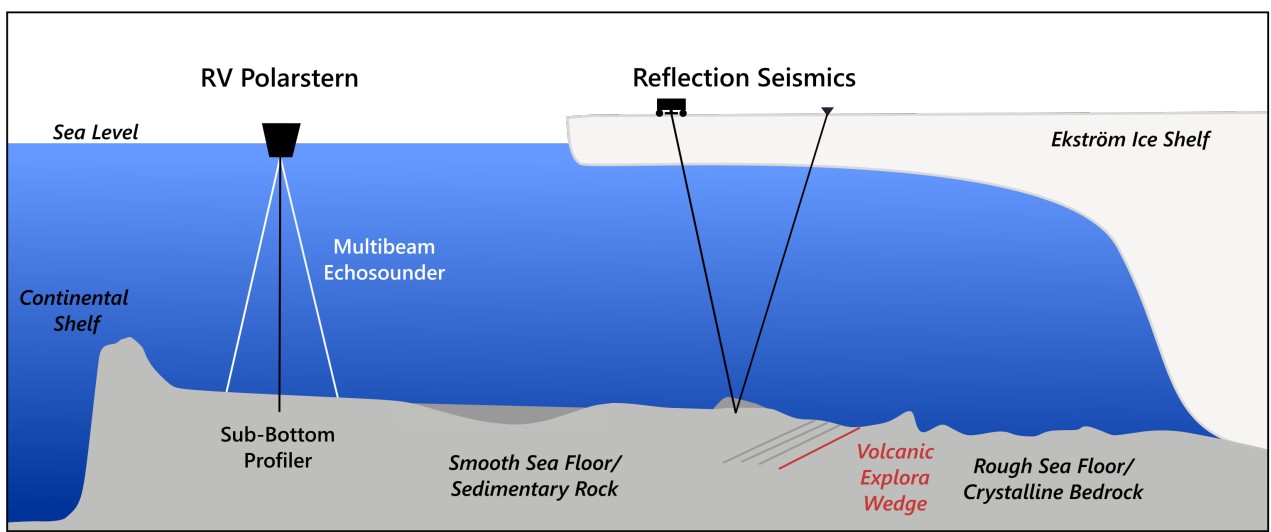

**Figure 2.** Schematic display of the study area and the used methods. The shown features are not to scale.



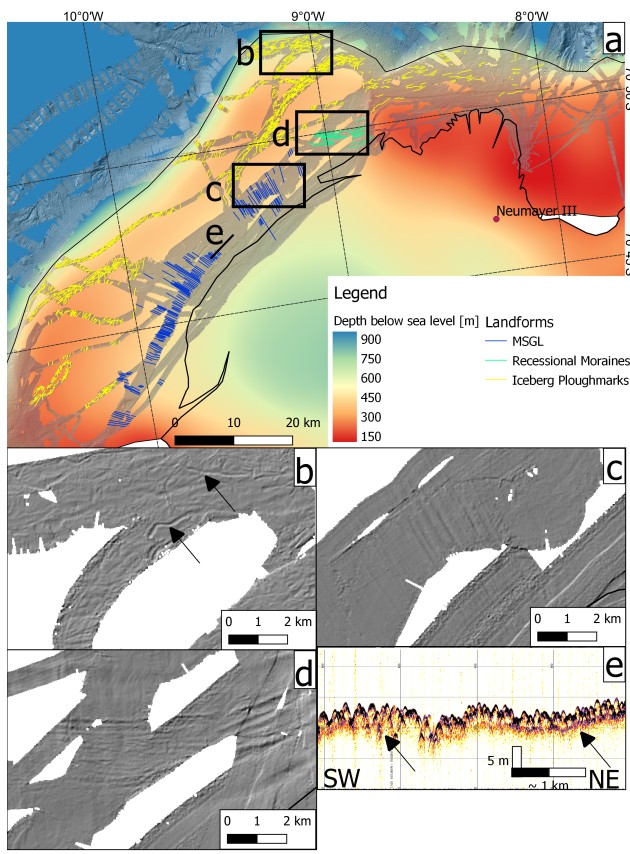

**Figure 3.** Landforms at the Continental Shelf. The panel (a) displays the water depth and an overview of the identified landforms. The rectangles in the bathymetry map (a) indicate the location of the close-ups of iceberg ploughmarks (b), MSGLs (c) and recessional moraines (d). Their orientation is the same as in panel (a). (e) is an echogram of the MSGL, the left black arrow indicates the absence of a basal layer, the right arrow its presence.

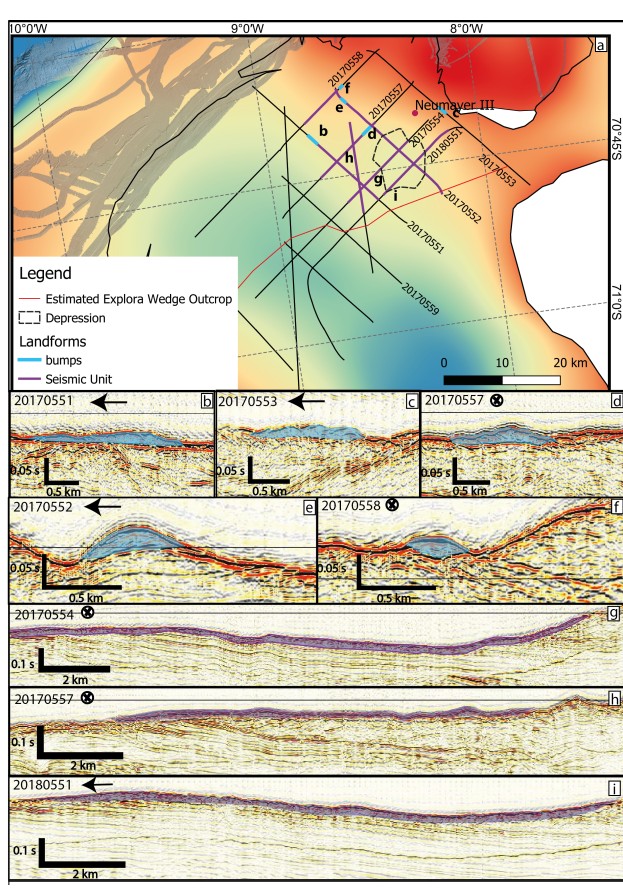

**Figure 4.** Landforms downstream of the Explora Wedge. (a) provides an overview of the found landforms. The panels (b)-(f) show the bumps in sediment, (g)-(i) display examples for the seismic unit interpreted as subglacial till. The arrow indicates the direction of the current ice flow direction with respect to the seismic line. The colour scale in (a) represents the depth below sea level as in Fig. 1.



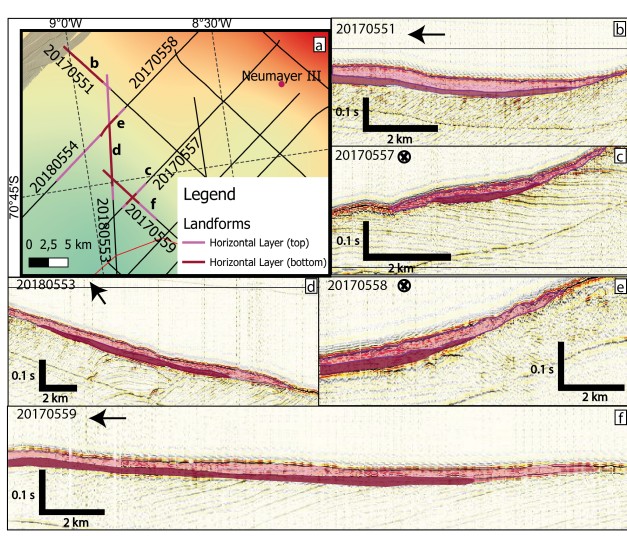

**Figure 5.** Panel (a) gives an overview of the location of the sea floor-parallel seismic unit. The panels b - f show the extent of the unit in the trough mouth. The dark pink represents the bottom layer, the light pink the layer on top. The arrow indicates the direction of the current ice flow direction with respect to the seismic line. The colour scale in (a) represents the depth below sea level as in Fig. 1.



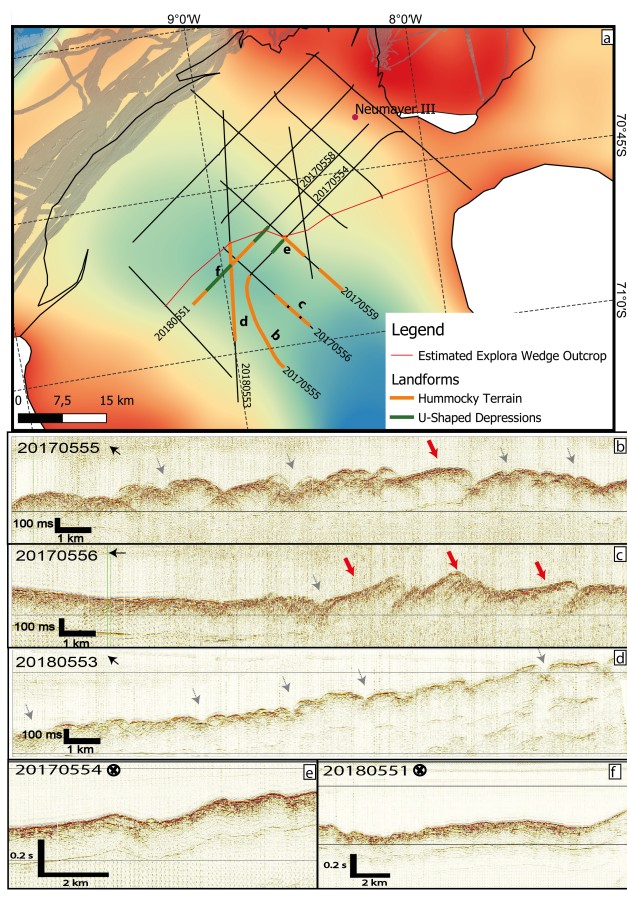

**Figure 6.** Landforms upstream of the Explora Wedge outcrop. The locations of the irregular sea floor and U-Shaped depressions are displayed in (a). Examples for the rough sea floor is shown in (b)-(d), the red arrows indicate the 'Hummocky Landforms in Bedrock'. The small depressions, which are pointed out with greyish dotted arrows, are seismic artefacts from crevasses in the ice shelf base above. A close-up of the U-Shaped depression in seismic line 20170554 is in panel (e), (f) displays these landforms in seismic profile 20180551. The colour scale represents the depth below sea level as in Fig. 1. The arrow indicates the direction of the current ice flow direction with respect to the seismic line.

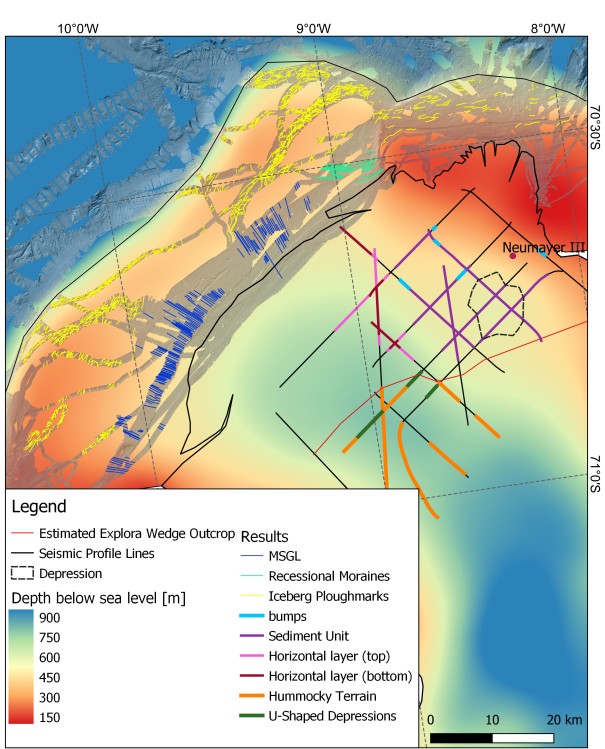

**Figure 7.** Overview map of the identified landforms. The length of the individual colored lines corresponds to the detected extent of the landform in the seismic data. The hydrographic line is marked in as thin black line in at the contintal shelf, the ice front is marked with a thick black line 25 km further south.