# Peer review of "Geomorphology and shallow sub-sea floor structures underneath the Ekström Ice Shelf, Antarctica"

_The Cryosphere, 2021_

## Author Response (AR1)

Referee 1:

Section 3.1: Worth mentioning broad survey layout in respect of total line length and an indication of spacing etc.

In case for the multibeam echosounder, the tracks are irregularly spaced, their width are variable (depending on the number of beams (see supplement)) and some of them lie on top of each other. Therefore, an indication of the total length is of limited informative value. Here, the important information is gridding, which is mentioned with 25 m resolution. The course of the cruises is shown in Fig. 1b.

Sentence in line 84 will be changed to: "The master tracks of these RV *Polarstern* expeditions, which have a total track length of ~ 400 km in the considered area, are shown in Figure 1b."

L90: Worth summarising the broad survey layout.

Sentence will be changed to: "The seismic profiles are orientated across or along the predominant ice-flow direction and shown in Figure 1b."

A more detailed survey layout was considered, but to avoid several repetitions (details of acquisition on the Ekström Ice Shelf are given in Smith et al. (2020), Fig 1b, Section 5.2 and "Assessment of sub-ice shelf seismic data for the interpretation of glacial landforms" (Line 305)), we decided to keep it short and refer to the named sections.

L110: you say they are randomly oriented, but you don't mention that they have variable orientations along their length too. Their along-length geometry is also an indicator that they are ploughmarks.

True. Sentence will be changed to: "These furrows are randomly oriented, changing orientation within their tracks and occasionally cross-cutting."

Section 4.1.2 Can the statistics (e.g. lengths, widths, orientations, elongation ratios) be shown graphically (e.g. distribution histograms etc.).

A statistical analysis of the measured MSGLs would be possible in principle, but it is evident (especially in the west of the study area) that the lengths of the MSGLs were not fully recorded with the multibeam echosounder (Figs. 3a and c). Overall, only a few MSGLs were imaged in their entire length. Less wide MSGLs can only be resolved with the sub-bottom profiler, which also imaged a very limited area of the overall study area (Figs. 1 and 3e). There, it can also be seen that the MSGLs partially overlap, which makes an exact determination of the widths difficult. Therefore, a statistical evaluation of the lengths, widths and elongation ratios would not be meaningful.

Therefore, we limit ourselves only to the indication of the predominant orientations.

L206: what it meant by short lived? And how is it known that they represent a short period?

Longer grounding line stillstands are associated with the formation of GZWs. Since there are no indications for a longer grounding line halt, we assume the stillstand to be relatively short, at least too short for a formation of GZWs.

L211: different origins? Could the ice origin actually be the same glacier but that the extent of that glacier may have been different (e.g. shorter and thus calving front was thicker) when the large icebergs were discharged. Feels too speculative that the origin was different. Or maybe not if the two sets are being created at the same time.

A bit more of an explanation is given in Line 214-219.

There is a bathymetric sill (with ~ 390 m water depth at its deepest point) at the continental shelf break, which separates the open ocean from the over-deepened incised trough. It acts as a barrier that prevents thicker icebergs coming from the ocean from entering the inner trough and vice versa. Thus, even if a thicker calving front had been present, icebergs from the Ekström Ice Shelf could hardly have crossed the bathymetric sill to leave ploughmarks seaward of the sill at water depths of ~535 m.

L232: How would it effect the basal thermal regime?

At the transition zone, there is a change between rough, crystalline bedrock and probably soft sediment. The smooth seafloor present downstream of the Explora wedge is assumed to reduce basal friction, favoring fast ice flow velocities and a warm-based thermal regime (e.g. Wellner et al., 2001; Ó Cofaigh et al., 2002; Lowe and Anderson, 2003; Siegert et al., 2005; Benn and Evans, 2010). More details on the respective paleo-thermal regime and ice flow regime is provided in the interpretation of the individual lanforms. A brief summary sentence is provided in line 368.

L240: What is meant by basal moraines?

Sentence will be changed to: "Basal tills may show MSGLs (e.g. Ó Cofaigh et al. (2002);Dowdeswell et al. (2004)), however, …"

L240-243 – this is speculation and I would remove it.

Sentence will be changed to: "Basal tills may show MSGLs (e.g., Ó Cofaigh et al. (2002); Dowdeswell et al. (2004)), but since the resolution and line spacing of the used seismic data are not sufficient to image MSGLs of the dimensions seen in front of the ice shelf, a presence of MSGLs below the shelf ice cannot be detected. "

L250-253: retreat and readvance are proposed, but no age determination is possible. I think it ought to be made clear that there is no information about the length of time between retreat and readvance. This is explored in the following paragraph a little and on

Sentence in L254 will be changed to: "An indication that periods of retreat and re-advance are possible in the EIS embayment is given by Schannwell et al. (2020), who simulated an idealised advance and retreat history of the grounding line of the EIS embayment over the past 40,000 years, however, there is no evidence to narrow down the timing between retreat and re-advance."

line 260 it needs to be made clear why this is 'less likely'. It also needs to be made clear that the Schannwell model does not simulate any readvance (yes a stillstand, but nothing more, despite the forcing). Indeed, I am not clear how changing the bed conditions would force a readvance. Could the authors reduce the speculation in this paragraph?

Paragraph will be changed to: "Schannwell et al. (2020) simulated an idealised advance and retreat history of the grounding line of the EIS embayment over the past 40,000 years in the EIS embayment. The simulations used a variety of bed conditions and showed that with both hard and soft beds, there were three periods of grounding line stillstand during advance and one in the retreat phase. However, the situation is likely more complex that that simulated by Schannwell et al. (2020), as our investigations have revealed a change in bed conditions 150 km from the grounding line, rather than a purely hard or soft bed, which has not been considered in the simulations so far. In general, short periods of re-advance after groundling line stillstand are conceivable at the EIS embayment, as ice sheets in the Weddell Sea are discussed to have undergone several re-advances (Davies et al. 2011 and references therein). An alternative scenario is that the lower layer originates from an earlier glaciation. However, this interpretation will remain ambiguous without age estimates from cores."

L270: compare also to the work of Ely et al 2016 – Geomorphology: Do subglacial bedforms comprise a size and shape continuum? - ScienceDirect) who explores geometries in detail including the transition from various elongated landform types.

Paragraph between L 267-271 will be changed to: "The more symmetrical shape of the bumps revealed by the across-flow profiles (Fig. 4d, f) may support an interpretation as drumlins. If we assume the along-flow and across-flow lines are cutting a family of such features orthogonally, then the heights and lengths of these features are within typical ranges for drumlinoids (Clark et al., 2009). Compared to Clark et al. (2009), the feature in Figure 4f is within the range for the typical width of a drumlinoid, however the landform in Figure 4d is wider than an expected width of 1500 m. However, compared to the work of Ely et al. (2006), who investigated the shape and size of subglacial bedforms in some locations of the northern hemisphere, the width of the feature in 4d fits to their reported width-range of 1510 m for drumlins. "

L340-343: The comment about expectations for similar behaviour in other troughs – I think that if we have learned anything from other regions it is that behaviour can vary significantly even between neighbouring troughs. This point therefore does not seem to be a safe one to make.

Generally yes, but we assume that the neighboring ice shelves do not behave fundamentally differently to the EIS. The paragraph will be changed to: "The EIS is a typical ice shelf of the wDML region (Neckel et al., 2012). Although it appears that adjacent ice shelves within Antarctica behave differently, the neighboring ice shelves of EIS have a similar narrow continental shelf and are as well underlain by incised troughs (Eisermann et al., 2020; Nøst, 2004; Favier et al., 2016), which may result in similar palaeoice thickness, retreat and advance styles. Relatively warm Circumpolar Deep Water masses have a significant influence on ice shelf variability (e.g. Jacobs et al., 1996, 2011; Hillenbrand et al., 2017; Paolo et al., 2015; Hattermann, 2018). However, as assumed for the neighbouring Fimbul Ice Shelf (Hattermann et al., 2014) as well as for the EIS (Smith et al., 2020), these warm water masses enter the cavity below the ice shelf only in relatively small amounts."

Technical corrections:

L37: Tending towards methodology – save for later. Also, what is meant by 'bottom' of the ice shelf? Underside?

We want to briefly touch on the possibility of investigating sub-ice shelf cavities with AUVs, but want to focus only on the methods actually used is the method section.

Yes, the 'bottom' of the ice shelf is the underside.

L39-41: Not sure this is particularly useful to the science being presented here.

It explains why we focus on the approach of combining reflection seismic data with marine geophysical methods as an alternative (in a broad sense) to expensive AUVs to image areas below ice shelves.

L60: semi colon between Smith and Eisermann references – should be an 'and'?

Sentence will be changed to: "Schannwell et al. (2020), using the bathymetry of Smith et al. (2020) and Eisermann et al. (2020), modelled the evolution…"

L205: 'fare' – doesn't seem right word. Provide?

Sentence will be changed to: "…fresh appearance of MSGLs and recessional moraines in the multibeam data, provide evidence that these features are relatively young,…"

L278: 'preferably' should be preferentially.

Sentence will be changed to: "...were found to be preferentially formed at the transition..."

L330: 'worth to be' should be 'worth applying'.

Sentence will be changed to: "...this methodology is worth applying and developed..."

L338: provide a reference or weblink to instant or don't mention. It's arguable that the instant programme itself need not be mentioned because that's not so relevant to the finding of the paper. The general point of this approach and work being useful can be made without the need to link to particular programmes.

Sentence will be changed to: "Our findings here can be used to determine and evaluate boundary conditions for palaeo-ice flow models of this wider region in this relatively poorly-investigated sector of East Antarctica, allowing us to understand and simulate future behaviour of the ice shelves more reliably."

L345-346: reference to the cruise next year. Be specific about the year. Also 'aimed to be' should be 'will be'.

Sentence will be changed to: "...will be collected during the *Polarstern* expedition EASI-1 in 2022 (Tiedemann and Müller, 2021)."

The cruise will be cited as: Tiedemann, R. and Müller, J. (2021). Expedition Programme PS128, Bremerhaven, Alfred Wegener Institute for Polar and Marine Research, 34 p. hdl:10013/epic.5f6c7ebb-86be-4277-a757-175baf6d916c

L351: Year for Patterson et al reference?

For Patterson et al. "under review" will be added.

L370: 'suggesting' should be 'suggest'.

Sentence will be changed to: "...which potentially suggest more intense meltwater flow..."

L380: 'to provide' should be 'for providing'.

Sentence will be changed to: "Hence, these methods are suitable for providing crucial constraints for palaeo-ice flow models..."

Figure 1: Why not write the PS1385 label onto the map? The grey contour line is not obviously grey – perhaps symbolise as a black dashed line?

'PS1385' will be added in Figure 1. The grey line will not necessarily be replaced by a black dashed line, as it could be confusing with the black dotted coordinate system, but it will be improved.

Figure 3: This figure needs to be full page width to enable clarity. Iceberg ploughmarks dash in the legend is difficult to see (yellow on white).

Figure will be changed to full page width. A contour line will be added in the legend.

Figures 4-7: these figures need to be full page width for clarity.

Figures will be changed to full page width.

Referee 2:

Abstract:

—MSGL end 11 km from shelf break indicating LGM position. That is minimum position but not necessarily max. Make this clear.

Sentence will be changed to: "Our surveys show that Mega-Scale Glacial Lineations cover most of the mouth of this trough, terminating 11 km away from the continental shelf break, indicating the most recent minimal extent of grounded ice in this region." Sentence in line 359 will be changed as well.

--What does beneath ~30 km of ice shelf mean? Is this the outer portion? Just make explicit.

This refers to the area below the ice shelf, from the ice shelf front/edge to ~30 km inland. The sentence will be changed to: "Beneath the front ~30 km of the ice shelf measured from the ice shelf edge towards inland direction, the sea floor is characterised by an acoustically transparent sedimentary unit,…"

--Till isn't usually 45 m thick. Is this really a single unit or multiple tills? If it is really 45 m, is that unique? Seems like a lot. (See below.)

It seems to be one unit in the seismic data. However, the resolution is not high enough to identify individual layers.

--Why does lack of over-printing indicate rapid retreat? It could be slow retreat. It simply indicates gradual, continuous retreat, with a lack of pauses along way. It doesn't have to be rapid.

Based on investigations by Shipp et al. (1999), Dowdeswell et al. (2004), Mosola & Anderson (2006) Cofaigh et al. (2008), good preserved MSGLs (e.g. at Marguerite Trough) and the absence of grounding-zone features a result of rapid ice sheet retreat.

Citation will be added to line 364.

Shipp, S., Anderson, J., & Domack, E. (1999). Late Pleistocene–Holocene retreat of the West Antarctic Ice-Sheet system in the Ross Sea: part 1—geophysical results. Geological Society of America Bulletin, 111(10), 1486-1516.

Dowdeswell, J. A., Ó Cofaigh, C., and Pudsey, C. J.: Thickness and extent of the subglacial till layer beneath an Antarctic paleo-ice stream, Geology, 32, 13–16, https://doi.org/10.1130/G19864.1, 2004.

Mosola, A. B., & Anderson, J. B. (2006). Expansion and rapid retreat of the West Antarctic Ice Sheet in eastern Ross Sea: possible consequence of over-extended ice streams?. Quaternary Science Reviews, 25(17-18), 2177-2196.

Cofaigh, C. Ó., & Stokes, C. R. (2008). Reconstructing ice-sheet dynamics from subglacial sediments and landforms: introduction and overview. Earth Surface Processes and Landforms: The Journal of the British Geomorphological Research Group, 33(4), 495-502.

Line 23-24: The sentence " Recent studies have shown that ice shelves in this region are underlain by deep glacial troughs (Nøst, 2004; Eisermann et al., 2020; Smith et al., 2020), indicating ice advanced towards outer continental shelf areas in the past." doesn't make sense. If the ice shelf has a trough under it, all it means is that grounded ice advanced as far as the modern-day ice shelf edge in the past. If showing that ice advanced towards the outer continental shelf in the past, need to say that the trough in fact extends that far, beyond the edge of the ice shelf.

The MSGLs prove, that grounded ice occurred maximal 11 km from the current calving front as well as 11 km from the continental shelf. The overdeepend trough beneath Ekström is assumed to be a result of former ice stream erosion (e.g. Smith et al. 2020). Therefore, the grounded ice extended shortly

before the bathymetric sill (see line 214). Floating ice, however, has probably reached across the continental shelf (Grobe and Mackensen, 1992, Smith et al. 2020).

Sentence will be changed to: "Recent studies have shown that ice shelves in this region are underlain by deep glacial troughs (Nøst, 2004; Eisermann et al., 2020; Smith et al., 2020), indicating that grounded ice advanced to the continental shelf in the past."

Line 34 (and related to the above): how narrow is the continental shelf? "Narrow" doesn't tell much.

The continental shelf is located 25 km from the current ice shelf front. As e.g. in Nøst, 2004, who as well describes the area at in the eastern Weddell Sea as narrow, we think an exact number does not really help to understand the geological setting. If interested, it can be measured in Fig. 1 and 7.

Nøst, O. A.: Measurements of ice thickness and seabed topography under the Fimbul Ice Shelf, Dronning Maud Land, Antarctica, Journal of Geophysical Research: Oceans, 109, https://doi.org/https://doi.org/10.1029/2004JC002277, 2004.

Line 48: "front" of the ice shelf is not especially clear. State seaward or from edge.

Sentence will be changed to: "Our study area covers the continental shelf and the sea floor under the first 50 km (measured from the calving front) of the EIS cavity (Fig. 1)."

Line 52: Not clear exactly what is being measured, from where to where. Annotate the figure and show these measurements so that it is explicit.

This makes the figure look rather chaotic than helpful. Therefore, we consider the description "…most landward location of the current grounding line (8°6'W at the southernmost point) to the continental shelf. The distance to the grounding line is measured along the trough axis." to be sufficiently clear.

Lines 57-58: Did the Grobe and Mackensen paper really claim that the ice extended beyond the continental shelf break? That would be quite extraordinary. I thought they looked at slope records and IRD, but not evidence of actual ice extending that far.

In the Grobe and Mackensen paper on page 373 'Glacial Maxima' as well as in Fig. 12 they state that grounded ice extended across the continental shelf break of East Antarctica. However, the sediment cores taken in the EIS region were too short to grab any datable material from the LGM.

Lines 60-70: In addition to the model estimates of LGM timing here, are there any estimates from dated cores?

The only chronological constraint is mentioned in Grobe and Mackensen, 1992, who set the age of their cores to the Holocene. Further dated sediment cores have, to my knowledge, not yet been published.

->4.1.2 and Figure 3: Certainly sounds like MSGLs that are being described. However, there is nothing convincing in the figure. Zoom in, show more, consider color, mark with arrows, etc. As it is, the figure is not convincing.

An overview of the found MSGLs are marked in blue in Fig. 3a. A zoomed-in section of the multibeam echosounder is displayed in Fig. 3c, and an example for the MSGLs in the echogram is shown in Fig. 3e. Since 3c and e show exclusively the MSGLs additional arrows were omitted.

->For comparison of how these sizes compare to other MSGLs, consider: Wellner, J. S., Heroy, D. C., & Anderson, J. B. (2006). The death mask of the Antarctic ice sheet: comparison of glacial geomorphic features across the continental shelf. Geomorphology, 75(1-2), 157-171. Or: Newton, M., Evans, D. J.,

Roberts, D. H., & Stokes, C. R. (2018). Bedrock megagrooves in glaciated terrain: A review. Earth-Science Reviews, 185, 57-79.

There are many MSGLs described in the literature, which allow a comparison with the found MSGLs at EIS. However, since our landforms can be identified unambiguously as MSGLs and are of typical dimensions, we do not aim to list detailed values for MSGLs of other study areas.

->For comparison of till and drape thicknesses, consider: Shipp, S., Anderson, J., & Domack, E. (1999). Late Pleistocene–Holocene retreat of the West Antarctic Ice-Sheet system in the Ross Sea: part 1—geophysical results. Geological Society of America Bulletin, 111(10), 1486-1516.

Sentences will be change to: "For example, studies on the Marguerite Trough show sediment thicknesses reaching 15 m (Ó Cofaigh et al., 2002) and 20 m (Dowdeswell et al., 2004), both similar to the minimal thicknesses seen in our study. In the Ross Sea, Shipp et al. (1999) describe a 2 m – 9 m thick transparent drape-unit and Karl et al. (1987) interpreted an up to 40 m thick overlying layer to be basal till.

Karl, H. A., E. Reimnitz, and B. D. Edwards. "Extent and nature of Ross Sea unconformity in the western Ross Sea, Antarctica." (1987).

->4.1.3: Many small moraines were noted in Antarctica much before the references cited.

Yes, a variety of different kinds of moraines are described in the literature. Here, publications were selected, due to the use of similar marine geophysical methods and study areas, in order to allow a more direct comparison with our identified landforms.

->For comparison of small-scale retreat features like moraines, consider: Shipp, S. S., Wellner, J. S., & Anderson, J. B. (2002). Retreat signature of a polar ice stream: subglacial geomorphic features and sediments from the Ross Sea, Antarctica. Geological Society, London, Special Publications, 203(1), 277-304.

Many different moraine types are discussed in Shipp et al. (2002). However, there is no unambiguous indication of retreat moraines in their zone 3 and 4. They identified retreat moraines in their Zone 5, but their dimensions are not given, which does not allow a sufficient comparison with our landforms.

Lines 201-202: Calling on a core that is not included in the data of the paper, nor apparently published elsewhere, is not a fair line of evidence.

We refer to the sediment core PS1385 (Figure 1 and Table 1), published in Grobe, H. and Mackensen, A.: Late Quaternary climatic cycles as recorded in sediments from the Antarctic continental margin, in: Antarctic Research Series, vol. 56, pp. 349–376, https://doi.org/10.1029/AR056p0349, 1992. They limit the age of the sediment core to the Holocene. The location of the core, where many iceberg plow marks are evident in the multibeam data, suggests that there is no longer undisturbed sediment deposition.

Lines 203-205: If evidence from slope cores indicates that ice reached shelf edge, but your MSGL data stops 11 km in and therefore you suggested ice maybe didn't reach the edge, how do those agree? Or, conflict?

The MSGL are replaced by the iceberg ploughmarks at about ~10 km from the current ice shelf edge. The iceberg ploughmarks were formed after the ice shelf retreated. They have overprinted the seafloor and destroyed potential MSGLs that no conclusions can be made about whether the MSGLs (and thus grounded ice) were present further toward the continental shelf.

Lines 236-240: It's a big stretch to say that your measurements of 45 m of till are "similar" in thickness to other places where it has been 15 m or 20m. That is more than 2x thicker, at least. Seems like a big difference! How does such a thickness of till work with the mode of formation of MSGLs? (See for example: Clark, C. D., Tulaczyk, S. M., Stokes, C. R., & Canals, M. (2003). A groove-ploughing theory for the production of megascale glacial lineations, and implications for ice-stream mechanics. Journal of Glaciology, 49(165), 240-256.) And, what does the statement about removal of loose sediments from under the ice by water have to do with the general idea of these being MSGLs?

Sentences will be changed to: "…show sediment thicknesses reaching 15 m (Ó Cofaigh et al., 2002) and 20 m (Dowdeswell et al., 2004), both similar to the minimal thicknesses seen in our study. In the Ross Sea, Shipp et al. (1999) describe a 2 m – 9 m thick transparent drape-unit and Karl et al. (1987) interpreted an up to 40 m thick overlying layer to be basal till."

Karl, H. A., E. Reimnitz, and B. D. Edwards. "Extent and nature of Ross Sea unconformity in the western Ross Sea, Antarctica." (1987).

MSGLs usually form under grounded ice in subglacial till. About the till thickness, on which the MSGLs are formed in the tough centre, we do not make any statement, since the marine geophysical data do not provide the needed in-depth data. In areas where the subglacial till deposits (with up to 45 m thickness) were detected with reflection seismics, MSGLs could not be resolved with our data.

The removal of loose sediments by water flow beneath the ice shelf does not question that these are MSGLs, but it might explain why rarely drape can be found on top of them.

FIGURES

Figure 1: Labels not clear in part a when text over ice boundaries. Define ice edge line in legend; differentiate grounding line from floating margin. Consider making this a three part figure as part a is currently not quite covering a large enough area to show people unfamiliar with Antarctica where the location is, but nor is zoomed in enough to show the region. Make an a and a b that are less/more zoomed in.

Labels and ice edge will be added. The white infill represents grounded ice, which will be clarified. A three-part figure will be made.

Figure 2: Very helpful schematic. Consider making more of the ice shelf under sea level and less of it above sea level. Also, consider moving the label of Ekstrom Ice Shelf out to the left a little, so that it is clearly on the ice shelf.

Figure will be changed, so that more of the ice shelf is below sea level and the label will be moved to the left.

Figure 7: Make it bigger! So much in there and way too hard to see!

Figure will be changed to full page width.

---

## Author Response (AR2)

Quantification of Items:

Narrow Shelf:

Sentence will be changed to: "The continental shelf in this area is with 30 km - 150 km particularly narrow (cf. Arndt et al., 2013) and is largely covered by thick ice shelves, preventing ship-based data acquisition."

Track width and line spacing (Section 3.1/Multibeam Echosounder):

Sentence will be changed to: "The data were recorded with the hull mounted multibeam echosounder systems Atlas Hydrosweep HSDS 2 and HSDS 3 (Knust, 2017). The tracks are partly overlapping or have spacings up to 7 km, the track widths range between 700 m and 2.3 km."

Line Spacing (Section 3.1/Sub-bottom Profiler):

Sentence will be changed to: "The master tracks of these RV Polarstern expeditions, which have a total track length of ~400 km in the considered area and partly overlap or have a distance of up to ~20 km, are shown in Fig 1c."

Grounding Ice on the Slope:

Following sentences will be deleted (Section 2 and Discussion): "Sediment core evidence offshore the EIS further suggests an extended East Antarctic Ice Sheet during marine isotope stages 6 and 4-2 probably reaching beyond the continental shelf break (Grobe and Mackensen, 1992), but no direct geomorphological or geological evidence exists for that so far (Hillenbrand et al., 2014)." And: "Other cores recovered from the continental slope suggest that grounded ice may have reached the shelf break during past glaciations, including the LGM (Grobe and Mackensen, 1992)."